# Impact of coronavirus disease 2019 on cancer care: How the pandemic has changed cancer utilization and expenditures

**Jinah Sim[1], Jihye Shin[2], Hyun Jeong Lee[3], Yeonseung Lee[3], Young Ae Kim[3]***

**1** School of AI Convergence, Hallym University, Chuncheon, Republic of Korea, **2** Department of Research, Health Insurance Review & Assessment Service (HIRA), Wonju, Republic of Korea, **3** Division of Cancer Control & Policy, National Cancer Center, Gyeonggi, Republic of Korea

☯ These authors contributed equally to this work.

* elkim7@gmail.com

## Abstract

### Purpose

Since identified in December 2019, the novel coronavirus disease 2019 (COVID-19) has had a global impact on medical resource use and costs for patients with cancer in South Korea. This study aimed to identify the medical use and costs among patients with cancer during the COVID-19 pandemic, to predict these patterns in South Korea in the future.

### Methods

We conducted a secondary claims data analysis using the National Health Insurance Service database for the calendar period of 2019–2020. Monthly relative percent changes in cancer incidence, medical use, and billing costs for medical care utilization by cancer type were calculated. Then, the medical use and costs after January 2020 were predicted using a time series model with data before the COVID-19 outbreak (2014–2019).

### Results

The incidence of cancer diagnoses has seen a notable decline since the outbreak of the COVID-19 in 2020 as compared to 2019. Despite the impact of COVID-19, there hasn't been a distinct decline in outpatient utilization when compared to inpatient utilization. While medical expenses for both inpatient and outpatient visits have slightly increased, the number of patients treated for cancer has decreased significantly compared to the previous year. In June 2020, overall outpatient costs experienced the highest increase (21.1%), while individual costs showed the most significant decrease (-4.9%) in June 2020. Finally, the number of hospitalisations and outpatient visits increased slightly from June–July in 2020, reducing the difference between the actual and predicted values. The decrease in the number of inpatient hospitalisations (-22~-6%) in 2020 was also high.

**Data Availability Statement:** The billing data from the National Health Insurance Service (NHIS) must undergo review by the NHIS Institutional Data Access / Ethics Committee after obtaining IRB

approval. Access to analyze the data is restricted to researchers who have been approved and meet the criteria set by the National Health Insurance Service (NHIS) for confidential data access. The data analysis takes place within a secure room at the public institution for approximately six months. Importantly, the data cannot be taken outside of this secured environment. Furthermore, only the analyzed results are permitted for external distribution. As a result, providing and accessing the data for journal purposes poses a considerable challenge. Contact details: URL: https://nhiss.nhis.or.kr/; E-mail: nhiss@nhis.or.kr; Phone: +82-1577-1000; Address: 32 Gungang-ro, Wonju-si, Gangwon-do 26464.

**Funding:** This study was funded by the National Cancer Center, Grant (No NCC 1911275 & 2210830-1 from Kim YA. The funders had no role in study design, data collection and analysis, decision to publish, or preparation of the manuscript.

**Competing interests:** We have read and understood your journal's policies, and we believe that neither the manuscript nor the study violates any of these. There are no conflicts of interest to declare.

**Abbreviations:** ARIMA, autoregressive integrated moving average; COVID-19, coronavirus disease 2019; NHIS, National Health Insurance Service.

## Conclusions

The overall use of medical services by patients with cancer decreased in 2020 compared with that in the pre-COVID-19 pandemic period. In the future, the government should consider how to recover from the COVID-19 pandemic, and establish permanent health policies for patients with cancer.

## Introduction

Since the first case of coronavirus disease 2019 (COVID-19) was identified in Wuhan in December 2019, the number of confirmed cases worldwide has surpassed 657 million, with 6.68 million deaths [1]. As of November 30, 2023, a total of 34,571,873 cumulative confirmed cases and 35,934 deaths have been reported in South Korea. Globally, it was reported to World Health Organization (WHO) that there are 772,052,752 confirmed cases of COVID-19 with 6,985,278 deaths [2]. According to a report by the WHO (2020), 42% of countries experienced cancer service disruption [3] owing to the pandemic; South Korea has also had negative changes in the medical use environment among patients with cancer [4, 5].

Patients with cancer are at high risk of infection owing to systemic immunosuppression caused by anticancer treatments such as chemotherapy or surgery [6], especially patients with active cancer, who are greater than five times more likely to die within 30 days than do patients in remission or with no evidence of cancer [7, 8]. In addition, patients with weak immunity or rare and severe incurable diseases may have avoided visiting medical institutions [9]. As a result, the decline and disruption in medical care systems have increased the risk of serious COVID-19 infection and death in patients with cancer [10].

Other than the direct medical impact from COVID-19 infections and associated mortality, the COVID-19 pandemic has led to widespread limitations in medical services [11]. Restrictions and isolation of hospitals may reduce the continuity of treatment in hospitals, leading to treatment delays or exclusions for serious complications or emergencies in patients with rare and severe incurable diseases. For example, a reduction in the Utilization of outpatient services, including cancer screenings, was observed internationally [12, 13], and the number of emergency department visits in South Korea significantly decreased [14]. Postponed medical use or changes in expenditure could increase the risk of delayed diagnosis and could be linked to more severe disease progression, which would have a marked indirect impact on long-term death rates [15].

Many studies have described the trends of the collateral damage or significant disruptions in medical use rather than forecasting the patterns of medical resource use and costs. Although some studies have estimated the changes in medical costs or use of resources related to COVID-19 using medical claims data [16], they did not consider the time-series use of medical resources and costs together. As COVID-19 is still highly contagious, health policies and systems should attempt to minimise those of damages from the COVID-19 pandemic, especially related to cancer care.

Awareness of the trends in medical use and costs before and after the COVID-19 pandemic can guide decision making regarding preventive interventions in healthcare delivery systems [17]. Therefore, we aimed to identify the patterns of medical resource use and costs among patients with cancer during the COVID-19 pandemic, to show these patterns in South Korea, using health claims data from the National Health Insurance Service (NHIS).

## Methods

### Database and patients

To estimate the impact of the COVID-19 pandemic on yearly economic costs and medical expenditures among the cancer population, we conducted a secondary claims data analysis using the NHIS, which includes anonymised information on insurance eligibility, treatments, and diagnostic procedures, and specifically diseases and prescriptions, from all patients who visited healthcare institutions nationwide. In our study, healthcare claims data with cancer codes (C00–C970 from 2014–2020) were collected. Patients who met the International Classification of Diseases 10th Revision Clinical Modification codes criteria, visited an outpatient clinic at least three times, and were hospitalised at least once within 1 year of a cancer diagnosis were included in the analysis [18]. This study was reviewed and approved by the Institutional Review Boards of the National Cancer Center (IRB No.: NCC2020-0087 and NCC2021-0015). The requirement for informed consent was waived owing to the retrospective nature of the study.

The cancer codes were classified into 24 groups as follows: lip, oral cavity, and pharynx (C00–C14); oesophagus (C15); stomach (C16); colon and rectum (C18–C20); liver (C22); gallbladder (C23–C24); pancreas (C25); larynx (C32); lung (C33–C34); breast (C50); cervix uteri (C53); corpus uteri (C54); ovary (C56); prostate (C61); testis (C62); kidney (C64); bladder (C67); brain and CNS (C70–C72); thyroid (C73); Hodgkin's lymphoma (C81); non-Hodgkin's lymphoma (C82–85, C96); multiple myeloma (C90); leukaemia (C91–C95); and other unspecified cancers. To calculate the costs and Utilization of cancer care, a period of exclusion from the patient's past history was required in the definition of cancer prevalence and incidence in this study.

### Outcome measures

This time-series analysis study identified whether changes in cancer care costs and utilization were affected by the COVID-19 pandemic. The NHIS claim database contains data on healthcare resource utilization, including disease diagnosis, medical treatment procedures, costs, and medication usage [19]. In our study, we calculated the resource utilization and costs, such as those related to administration, procedures, medication, injections, and other aspects of care, for the cancer population. We classified these into the categories of hospitalization and outpatient visit. Only the coverage by the National Health Insurance and co-payment with patients were considered. Specially for inpatient, the cost of admission was also included. If a patient had no medical resource utilization related to cancer reported in the prior year, medical resource utilization in the current year was defined as a new incidence of cancer. Finally, the amount of change in medical costs and utilization after January 2020 was calculated using data on medical utilization (i.e., the number of claims) before the COVID-19 pandemic. Monthly medical utilization and expenditures were calculated using detailed information from hospitalisation and outpatient treatment records from January 2014–December 2020.

### Statistical analysis

First, we calculated the relative percent changes (%) in cancer incidence, medical use, and billing costs for medical care Utilization by cancer type during the calendar period of 2019–2020 for each month. A washout period of 4 years prior to the year of analysis was established to calculate the cancer incidence. Changes in medical costs were then stratified by sex (S1 Fig), age (S2 Fig), cancer type (S3 Fig), and income level (S4 Fig). Finally, the medical use and costs after January 2020 were predicted using a time series model (autoregressive integrated moving

average [ARIMA] model) with data on medical use and costs (based on claims data) from before the COVID-19 outbreak (January 2014–December 2019). The predicted value from January–December 2020 obtained in the time-series model was compared with the actual medical use, and the rate of medical use related to COVID-19 was quantitatively presented. ARIMA models were used to predict future trends by incorporating the data characteristics (e.g., trends, seasonal information, cyclical information, and irregular variability) over a certain period of time on the premise that the trends and patterns of the past will remain constant. Since the health insurance claims data indicate specific periods in which the number of claims suddenly increases (i.e., every January and December) or decreases (i.e., every September), a seasonal time series model was also applied in this study. The results were further stratified by sex (S5 Fig), age (S6 Fig), cancer type (S7 and S8 Figs), and expenditures from admission and outpatient visit claims data (S9 and S10 Figs).

## Results

Overall, the incidence of cancer-related insurance claims decreased during the COVID-19 pandemic. In particular, during the period of intensive social distancing in March and April of 2020 versus 2019, the incidence of gastric and colorectal cancers declined sharply, which seems to be related to the decreased cancer screening rate. In 2020, compared to the same month in 2019, the incidence of cancer decreased overall, excluding lip, oral, and pharyngeal cancer (1.9%), pancreatic cancer (0.3%), and testicular cancer (2.9%). Particularly, there was a significant decrease in thyroid cancer (-11.9%), stomach cancer (-11.1%), and brain and central nervous system cancers (-10.9%) (Fig 1). Regarding the changes in medical use related to hospitalisation, the number of hospitalisations decreased during the COVID-19 pandemic. In particular, the number of hospitalisations has continued to decline since the high-intensity

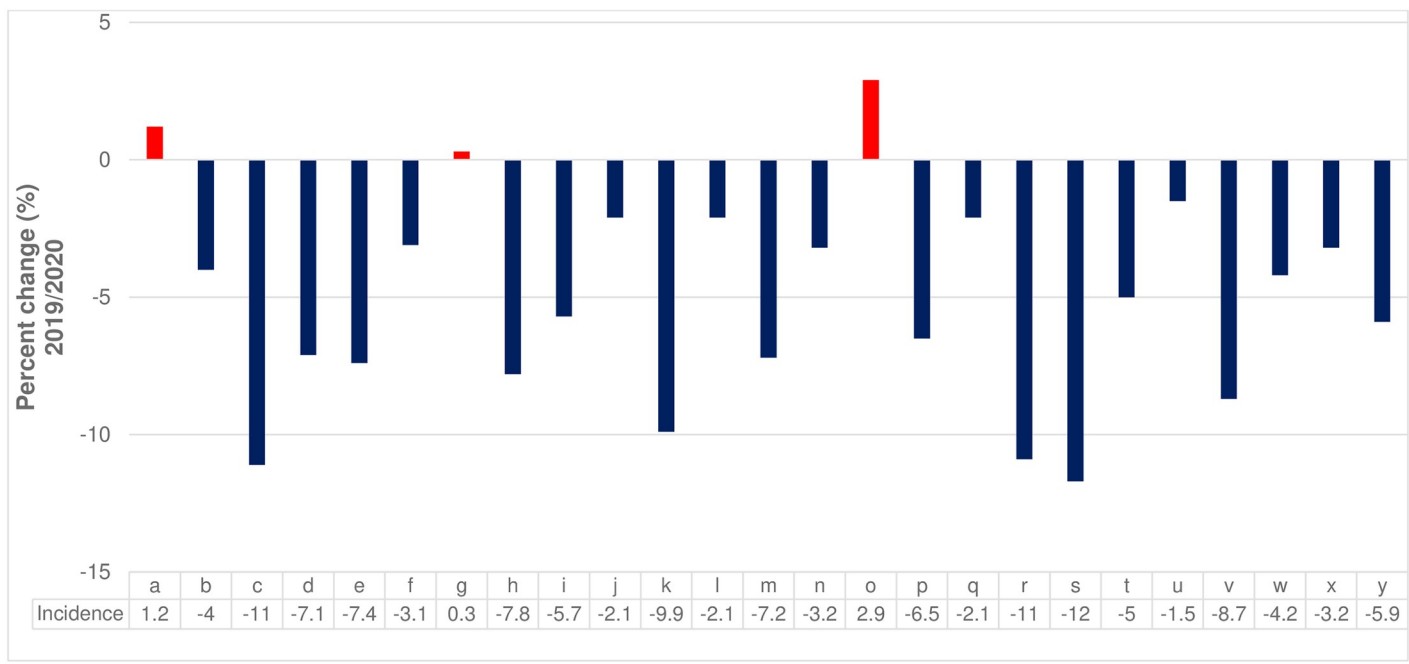

**Fig 1. Relative percent change (%) in cancer incidence by cancer type (2019 vs 2020).** a. lip, oral cavity, and pharynx; b. oesophageal; c. stomach; d. colon; e. liver; f. gallbladder; g. pancreatic; h. larynx; i. lung; j. breast; k. cervix; l. uterine cervical; m. ovarian; n. prostate; o. testicle; p. kidney; q. bladder; r. brain and central nervous system; s. thyroid; t. Hodgkin lymphoma; u. non-Hodgkin lymphoma; v. multiple myeloma; w. leukaemia; x. other; and y. all cancer.

social distancing policy was implemented (March–May) and during the second wave of the pandemic that occurred in a metropolitan area (August and September). In June 2020, outpatient visits (14.6%) showed a notable increase compared to the previous year, indicating a higher growth rate than in other months. Owing to the influence of COVID-19, the implementation of high-intensity social distancing seemed to have affected the number of outpatient visits, which decreased from April to May (ranging from -13.4% to -12.6%), and from August to October (ranging from -14.0% to -11.9%) (Fig 2).

The relative percent changes in billing costs related to medical use in 2020 compared with that in 2019 are presented in Fig 3. After the initial outbreak of COVID-19 in 2020, the number of hospital admissions among cancer patients decreased overall compared with that in 2019 (ranging from -15.5% to -2.7%), although the total hospitalisation costs increased slightly. During the first, second, and third waves of the pandemic, the total costs from admissions decreased slightly, although the individual costs from admissions increased compared with that in 2019. Since the COVID-19 outbreak in 2020, the total costs related to outpatient visits increased compared with that in 2019. Individual costs from outpatient visits also increased overall compared with those in 2019, and the total outpatient costs increased the most in June (21.1%) when the number of confirmed COVID-19 cases stabilised; in contrast, individual costs from hospitalized patients decreased the most in June (-4.9%).

After the first wave of the pandemic occurred in Daegu and Gyeongbuk from February–March 2020 in Korea, the proportion of patients hospitalised within 30 days of a cancer diagnosis showed the largest decrease (Fig 4). Breast (-3.9%), lung (-4.4%), and pancreatic (-4.6%) cancer patients showed the lowest decrease in the number of hospitalizations within 30 days compared to patients with other types of cancer. The reduction in the number of hospitalizations within 30 days for colorectal (-9.0%), stomach (-11.0%), and thyroid cancer (-11.7%) was particularly pronounced in July and August, which coincided with the period of the lowest social distancing measures due to the COVID-19 pandemic, compared to the same months in the previous year. Since then, the number of patients hospitalised within 30 days of a cancer diagnosis decreased in Seoul and Gyeonggi Province more than any other area owing to the

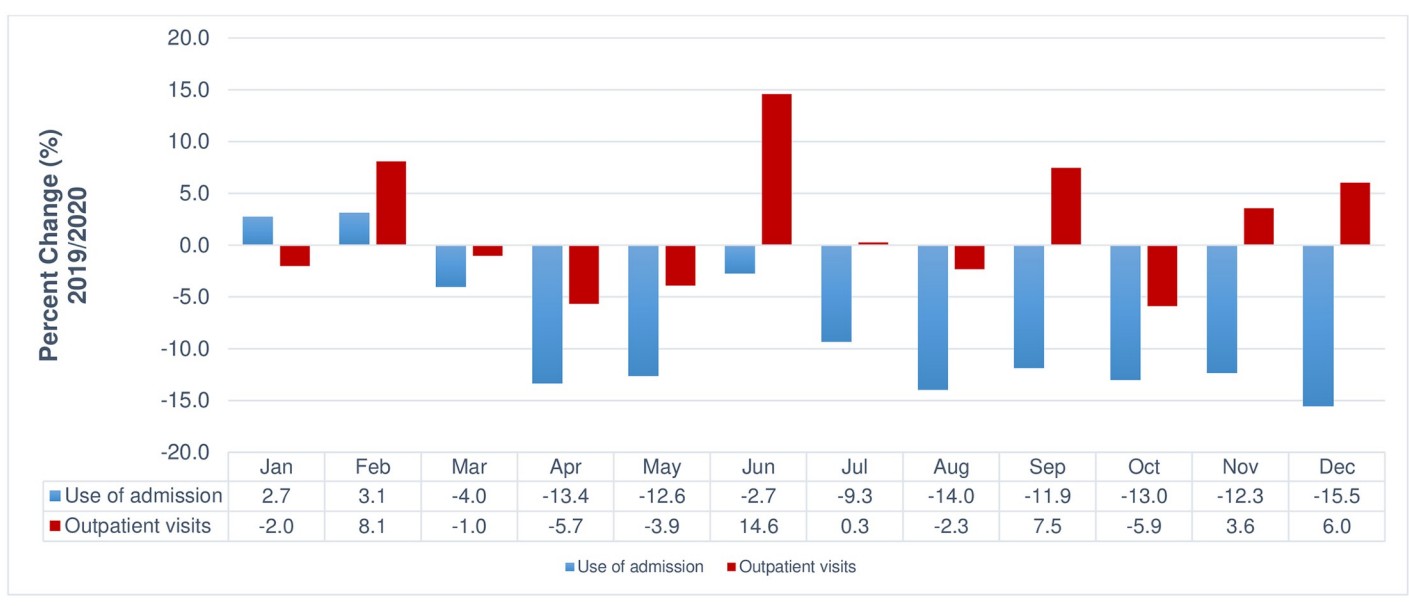

**Fig 2. Relative percent change in billing frequency for medical use (2019 vs 2020).**

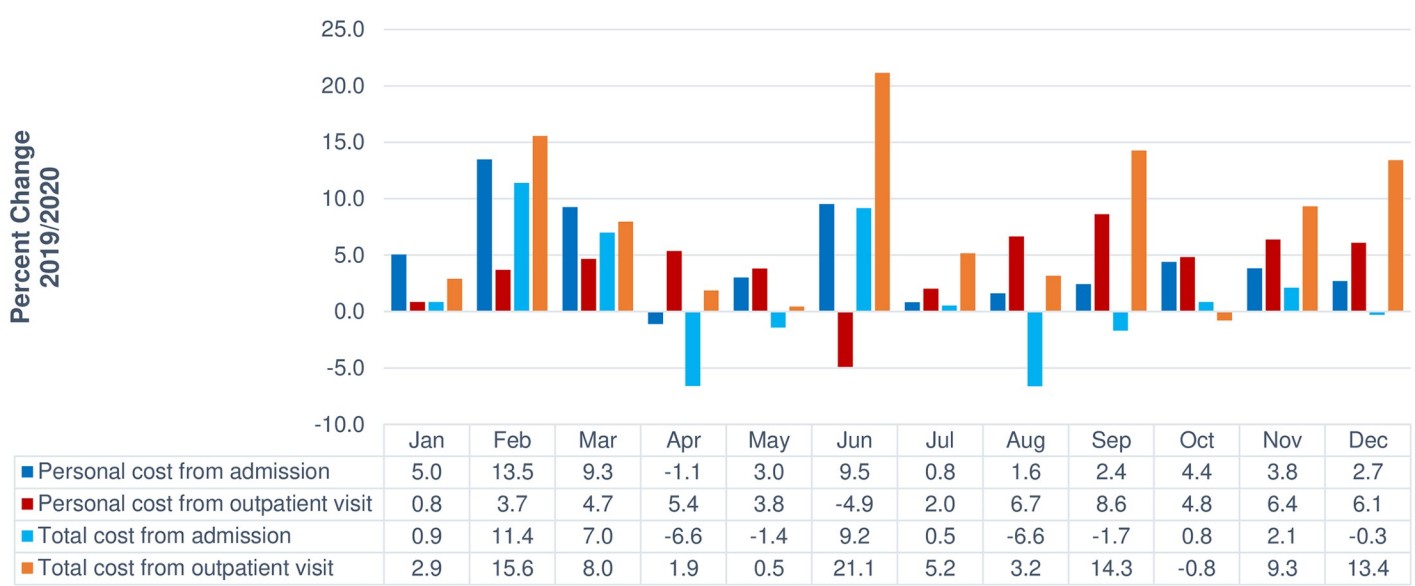

**Fig 3. Relative percent change in billing charges related to medical use (2019 vs 2020).**

second wave of the pandemic, which was centred in the Seoul metropolitan area, from July–August. Since November 2020, the number of patients with confirmed COVID-19 has increased to an average of 1,000 per day, and the number of inpatients nationwide has declined.

The number of medical claims for hospitalisations of cancer-related diseases showed an increasing trend from 2014–2019, before the COVID-19 outbreak of in Korea. However, the number of medical claims in 2020 decreased by 8.7% to approximately 1.42 million (S1 Table).

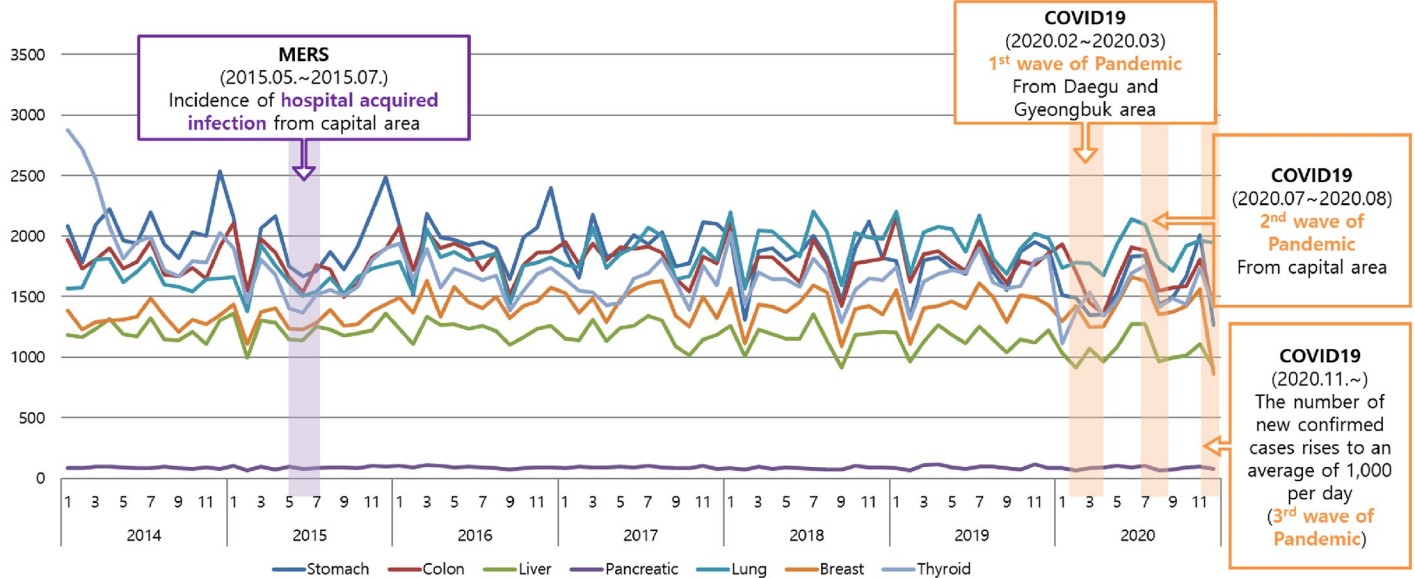

**Fig 4. Number of admissions within 30 days by cancer type (2019 vs 2020).** COVID-19, coronavirus disease 2019; MERS, Middle East respiratory syndrome.

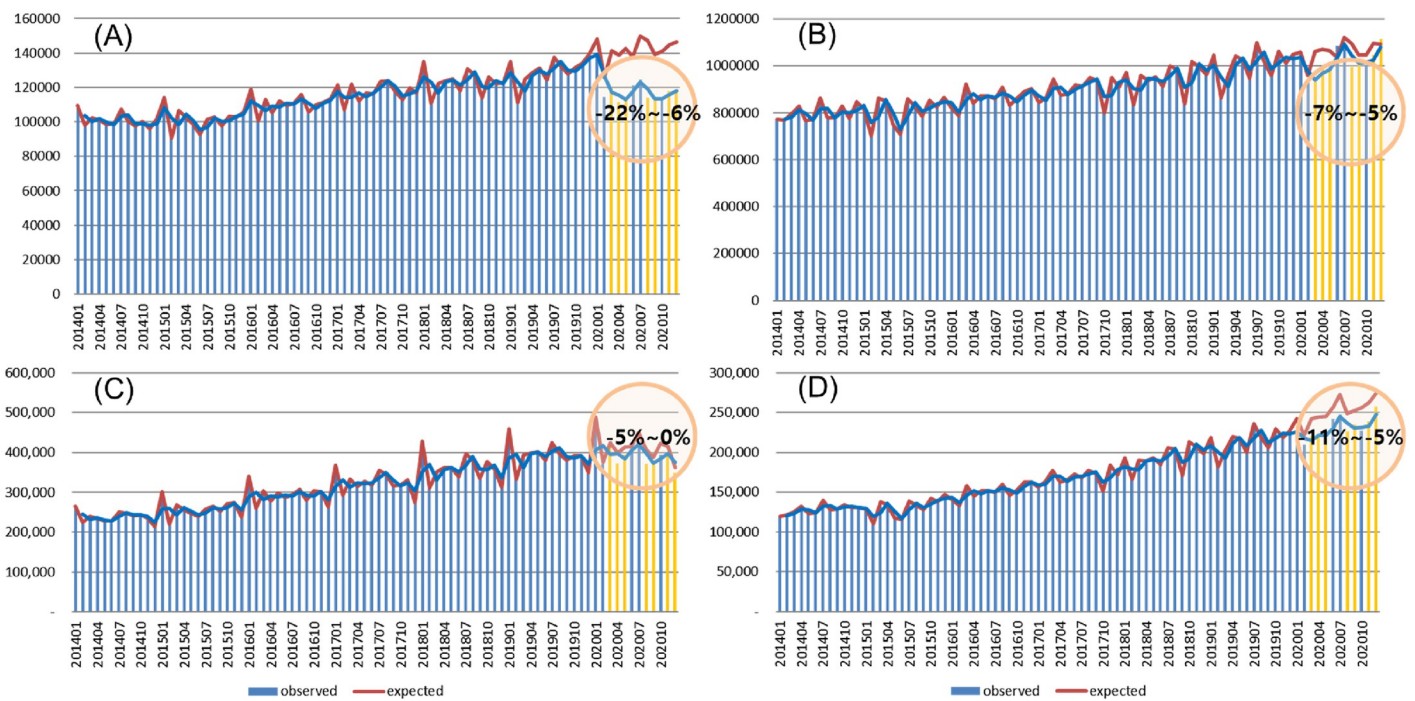

**Fig 5. Forecast of medical resource use and expenditures among cancer patients (2019 vs 2020).** (A) Number of admissions. (B) number of outpatient visits. (C) costs from admissions. (D) costs from outpatient visits.

The number of admissions and outpatient visits gradually increased from 2014–2019; however, the number of admissions and outpatient has decreased since February 2020 (Fig 5). The actual number of treatments compared with the number of treatments predicted using the seasonal ARIMA model and, in particular, the decrease in the number of inpatient hospitalisations (-22~-6%) compared with the number of outpatient visits (-7%~-5%) was higher. Owing to the COVID-19 pandemic, the actual number of treatments decreased significantly compared with the predicted number of treatments, although there was no significant difference in medical costs. Both the number of admissions and outpatient visits showed the largest difference between the actual and predicted number of cases during the first wave of the pandemic (February–March). The number of hospitalisations and outpatient visits increased slightly from June–July, reducing the difference between the actual and predicted values.

## Discussion

Using secondary claims data from January 2014–December 2020, this empirical study identified the changes in medical utilization and costs among cancer patients due to the COVID-19 pandemic. In addition, the use of medical resources and costs among cancer patients from January 2020–December 2020 were predicted using pre-COVID-19 outbreak data (i.e., January 2014–December 2019). Then, the actual changes in medical use were compared and analysed. Since the outbreak of COVID-19, the number of diagnoses has decreased significantly compared with that in the same period in the previous year. This decline particularly accelerated during the COVID-19 pandemic, and the decrease in the number of inpatient visits compared with outpatient visits was evident regardless of sex, age group, cancer type, or income level.

Although total medical expenses did not decrease significantly compared with the previous year, the number of patients who were treated decreased significantly.

Since the first COVID-19 outbreak in Korea on January 20, 2020, there have been three waves in Korea, (from February–March, August–September, and October–December 2020, respectively) [20–23]. However, owing to COVID-19, the established healthcare system was insufficient for patients with cancer. Therefore, the Korean government attempted to maintain national health strategies to control COVID-19 without adopting lockdown measures. During the COVID-19 pandemic, the government developed prompt and accurate diagnostic strategies and strengthened community-level preventive measures [21, 24, 25]. As cancer is a particularly severe disease globally [26], proper and continued treatment is important, and can be a critical factor in cancer mortality [27]. More recent studies indicate that if patients with cancer are infected with COVID-19, they are more likely to die within 30 days compared with individuals without any evidence of cancer [28]. Therefore, policies regarding cancer treatment and management are crucial for patients who might have missed some treatments due to the COVID-19 pandemic [4, 28–30].

A major finding of the present study was that overall medical utilization regarding cancer in 2020 decreased compared with that in the pre-COVID-19 pandemic period. In particular, the number of admissions decreased significantly compared with that in the pre-COVID-19 pandemic period, especially when the pandemic spread throughout certain regions [31]. These study findings are consistent with those reported by other studies from South Korea that analysed large tertiary hospital data and found that cancer incidence, medical use, and number of admissions decreased slightly immediately after the first COVID-19 wave [14, 32]. In addition, the actual number of inpatient visits decreased significantly compared with the predicted value in the time series analysis owing to the influence of the pandemic. Similar to other countries [4, 33–35], the number of outpatient visits in Korea decreased significantly. The actual medical use of patients diagnosed with cancer may have decreased because of high levels of social distancing and behavioural changes (e.g., wearing a mask, hand hygiene) [36]. However, patients with pancreatic cancer had the lowest decrease in outpatient visits and hospitalizations compared to patients with other types of cancer. The study findings are justifiable, given the imperative nature of pancreatic cancer, which represents a major urgency in oncology and necessitates a substantial allocation of medical resources [37].

Our findings are consistent with those of Korean government reports released at the end of 2022. Compared with the averages for the same months from 2017–2019, the total number of cancer cases decreased in 2020 by 18.7% and 14.4% in March and April, respectively, when the first COVID-19 pandemic wave occurred, and high-intensity social distancing was implemented, although they increased by 10.7% in June when the number of confirmed cases stabilised. Results of medical use and costs were also similar to this incidence pattern. The number of patients with cancer has steadily increased over the past 5 years (2016–2020) although the number of new patients with cancer has decreased by 3.0% over the past year. Moreover, compared with the averages for the same months from 2017–2019, the number of hospitalised patients and outpatients decreased in 2020 by 16.4% and 16.5% in March and April, respectively. This further decreased by 3.0% in June and July before gradually increasing to 13.8% and 14.1%, respectively.

The cancer screening rate and number of patients undergoing cancer diagnostic tests also decreased. For example, a breast cancer simulation model developed by the Cancer Intervention and Surveillance Modeling Network predicted the impact of the COVID-19 pandemic on breast cancer screening and of delayed diagnosis on breast cancer mortality from 2020–2030. Therefore, rapid cancer screening and efforts to minimise delays in diagnosis for patients with symptoms should be maintained to significantly mitigate pandemic-related impacts [38].

According to Massachusetts General Brigham Healthcare System patient records, the number of cancer screenings and patients diagnosed during the first pandemic wave were similar; however, several screening test rates were significantly decreased, along with the diagnosis rate of patients with cancer [39]. According to Korea Central Cancer Registry, there was a decrease in the number of cancer patients in 2020, but an increase was observed in 2021 (The results will be released at the end of December 2023.). This suggests a registration delay, indicating that the decrease in cancer patients in 2020 might have been influenced by COVID-19, even if the real incidence of cancer did not decrease. Indeed, especially during the period of high-intensity social distancing, there was a significant decrease in the incidence of stomach, colorectal, liver cancer, and breast cancer. Considering that stomach, liver, colorectal, breast, cervical, and lung cancers, can be screened through the Korean National Cancer Screening Program [40], it can be attributed largely to the significant decrease in cancer screening rates during the COVID-19 pandemic [5, 41].

Meanwhile, medical costs showed a slight increase for both hospital admissions and outpatient visits. As the COVID-19 pandemic rapidly spread worldwide, many experts insisted that medical costs would rise substantially due to COVID-19 [42–44]. In the United States, several studies estimated the immediate or direct medical costs of COVID-19 and predicted that there would be high costs related to hospitalisations [16, 45–47]. In contrast, some experts insisted that medical costs seemed to be substantially reduced; ultimately, the COVID-19 pandemic resulted in economic burdens for patients with certain diseases or older patients [48]. Considering our findings that medical costs increased inversely with the significant decrease in the number of admissions, the financial loss due to COVID-19 has been immense and has threatened the financial stability of vulnerable individuals [46, 47, 49]. In addition, outpatient medical expenses for pancreatic, lung, breast, liver, and other types of cancer increased compared to the same period the previous year. While in terms of inpatient costs, during the period of intensified social distancing measures, there was a significant decrease in inpatient costs for patients with most cancers, excluding pancreatic cancer. Furthermore, the government should consider how it may struggle to recover from the COVID-19 pandemic and establish permanent health policies for patients with cancer considering their cancer type and their vulnerability.

The observed trends of our study results underscore the urgency of addressing the pandemic's impact on cancer care and emphasize the need for strategic recovery efforts within the South Korean healthcare system, particularly to the challenges faced by cancer patients. Therefore, it is anticipated that the South Korean government will soon discuss the final outcomes of efforts to terminate COVID-19. The government plans to make statements on both COVID-19 and cancer after the official announcement of cancer registration statistics for the years 2022–2023, scheduled for the latter half of next year. Currently, South Korea is implementing 'essential healthcare' policies in response to a shortage of specialized personnel for COVID-19. The primary goal of 'essential healthcare' is to emphasize the accessibility of medical services in residential areas [49, 50]. This emphasis has been particularly crucial in regions where healthcare utilization was significantly impacted during the period of epidemic spread, especially evident in certain areas, consistent with our results. Taking into account the substantial decrease in healthcare utilization during the epidemic's spread, particularly in certain regions, and the emergence of regional disparities in COVID-19, it is expected that this policy will prove highly beneficial.

Our study had some limitations. First, because the use of medical services in our study depended on claims data, identifying the actual use was not possible unless the medical institution submitted a claim for medical expenses. More than 95% of claims were completed within 3–6 months after December 2020; therefore, some claims may not be included. Second, the

medical use prediction model was established using data on secondary claims of cancer patients, and the decrease was calculated based on the actual values. To establish a predictive model for the medical use of patients with cancer, the accuracy of the predictive model must be increased, and further detailed analyses should be conducted considering various factors that affect medical use. Third, the NHIS data lacks information on cancer stage, investigating the impact of COVID-19 across different cancer stages poses a challenge. In addition, presenting prompt statistical analyses to clarify the association between the observed trend and COVID-19 is challenging. Due to the urgency of our research, we relied on National Health Insurance data, encompassing the entire Korean population. However, persistent challenges in accessing NHIS data expected only after 2025, coupled with difficulties arising from delayed cancer registration announcements, make it difficult to conduct additional statistical analyses.

In conclusion, our findings indicate that the overall use of medical services by patients with cancer in 2020 decreased compared with that in the pre-COVID-19 pandemic period. In particular, the number of hospitalisations decreased significantly compared with that before the pandemic. Medical expenses increased slightly for both hospitalisations and outpatient visits. Notably, medical expenses increased despite the significant decrease in the number of hospitalisations. Findings from this work will generate insights about the impacts of the COVID-19, and inform future preparedness.

## Supporting information

**S1 Table. Comparisons of average of admission, average of admission days per person.** (DOCX)

**S1 Fig. The effect of utilization due the characters of cancer patients (sex).** (TIF)

**S2 Fig. The effect of utilization due the characters of cancer patients (age).** (TIF)

**S3 Fig. The effect of utilization due the characters of cancer patients (cancer type).** (TIF)

**S4 Fig. The effect of utilization due the characters of cancer patients (income).** (TIF)

**S5 Fig. Forecasts of cancer patients' medical use (sex/ admission cases).** (TIF)

**S6 Fig. Forecasts of cancer patients' medical use (age/admission cases).** (A) Under 30s, (B) 30s, (C) 40s, (D) 50s, (E) 60s, (F) 70s, (G) 80s. (TIF)

**S7 Fig. Forecasts of cancer patients' medical use (cancer type/cases).** (A) Stomach admission, (B) Colon admission, (C) Liver admission, (D) Pancreatic admission, (E) Lung admission. (TIF)

**S8 Fig. Forecasts of cancer patients' medical use (cancer type/cases).** (A) Breast admission, (B) Thyroid admission, (C) Pancreatic outpatient, (D) Lung outpatient. (TIF)

**S9 Fig. Forecasts of cancer patients' medical use and expenditure (income/admission cases).** (A) Total medical benefit, (B) 1st Quartile, (C) 2nd Quartile, (D) 3rd Quartile, (E) 4th

Quartile, (F) 5th Quartile.
(TIF)

**S10 Fig. Forecasts of cancer patients' medical use and expenditure (income/outpatient cases).** (A) Total medical benefit, (B) 1st Quartile, (C) 2nd Quartile, (D) 3th Quartile, (E) 4th Quartile, (F) 5th Quartile.
(TIF)

## Author Contributions

**Conceptualization:** Jinah Sim, Jihye Shin, Hyun Jeong Lee, Yeonseung Lee.

**Data curation:** Jihye Shin, Hyun Jeong Lee, Yeonseung Lee, Young Ae Kim.

**Formal analysis:** Jihye Shin, Yeonseung Lee, Young Ae Kim.

**Funding acquisition:** Young Ae Kim.

**Investigation:** Jinah Sim, Jihye Shin, Young Ae Kim.

**Methodology:** Jinah Sim, Jihye Shin, Young Ae Kim.

**Project administration:** Young Ae Kim.

**Resources:** Young Ae Kim.

**Supervision:** Young Ae Kim.

**Visualization:** Jinah Sim, Jihye Shin.

**Writing – original draft:** Jinah Sim, Jihye Shin, Hyun Jeong Lee, Yeonseung Lee, Young Ae Kim.

**Writing – review & editing:** Jinah Sim, Jihye Shin, Hyun Jeong Lee, Yeonseung Lee, Young Ae Kim.

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
