## [Decision Letter · Decision Letter 0]

23 Oct 2023

PONE-D-23-21127Impact of Coronavirus Disease 2019 on Cancer Care: How the Pandemic has Changed Cancer Utilization and ExpendituresPLOS ONE

Dear Dr. Kim,

Thank you for submitting your manuscript to PLOS ONE. After careful consideration, we feel that it has merit but does not fully meet PLOS ONE’s publication criteria as it currently stands. Therefore, we invite you to submit a revised version of the manuscript that addresses the points raised during the review process.

Please revise your article according to the reviewers' suggestions listed below. 

We look forward to receiving your revised manuscript.

Kind regards,

Chong-Chi Chiu

Academic Editor

PLOS ONE

Journal Requirements:

https://www.e-crt.org/m/journal/view.php?number=3081

In your revision ensure you cite all your sources (including your own works), and quote or rephrase any duplicated text outside the methods section. Further consideration is dependent on these concerns being addressed.

"This study was funded by the National Cancer Center, Grant (No NCC 1911275 & 2210830-1 from Kim YA."

4. Thank you for stating the following in the Competing Interests: 

"This work was supported by the National Cancer Center [NCC 1911275, 2210830-1]. "

We note that one or more of the authors have an affiliation to the commercial funders of this research study : National Cancer Center 

Within your Competing Interests Statement, please confirm that this commercial affiliation does not alter your adherence to all PLOS ONE policies on sharing data and materials by including the following statement: ""This does not alter our adherence to  PLOS ONE policies on sharing data and materials.” (as detailed online in our guide for authors http://journals.plos.org/plosone/s/competing-interests). If this adherence statement is not accurate and  there are restrictions on sharing of data and/or materials, please state these. Please note that we cannot proceed with consideration of your article until this information has been declared.

Additional Editor Comments:

Please revise your article according to the suggestions of two reviewers.

Reviewers' comments:

Reviewer's Responses to Questions

**Comments to the Author**

1. Is the manuscript technically sound, and do the data support the conclusions?

Reviewer #1: Yes

Reviewer #2: Yes

2. Has the statistical analysis been performed appropriately and rigorously? 

Reviewer #1: I Don't Know

Reviewer #2: I Don't Know

3. Have the authors made all data underlying the findings in their manuscript fully available?

Reviewer #1: Yes

Reviewer #2: Yes

4. Is the manuscript presented in an intelligible fashion and written in standard English?

Reviewer #1: Yes

Reviewer #2: Yes

5. Review Comments to the Author

Reviewer #1: This study assessed the impact of COVID-19 on the diagnosis and management of patients with cancer. The authors found that the overall use of medical services by patients with cancer decreased in 2020 compared with that in the pre-COVID-19 pandemic period. This topic is interesting, and the manuscript is well-written. I just have several minor suggestions.

1. Add more detail data in the result part of abstract section and also result section.

2. Please update the epidemiology of COVID-19 in the introduction section.

3. Please define how to calculate the cost and medical resource utilization of patients with cancer in the method section.

4. May add statistical analysis to clarify the association between the trend and COVID-19.

5. Did this database include cancer stage? I wonder whether the effect of COVID-19 differ according to the cancer stage.

6. Please add the discussion about why the effect varied according to different cancer type.

Reviewer #2: This study investigated the impact of the COVID-19 pandemic on medical resource utilization and costs among cancer patients in South Korea. Using claims data, it observed a significant reduction in cancer diagnoses in 2020 compared to 2019, particularly in inpatient visits. While medical expenses remained stable, there was a noticeable decline in the number of cancer patients undergoing treatment. The study highlights the necessity for long-term health policies to address the pandemic's impact on cancer care and emphasizes the importance of future recovery efforts within the South Korean healthcare system for cancer patients.

Minor Suggestions:

1. Consider adding the cancer stage.

2. Include information about the outcomes of these cancer patients.

3. Discuss potential government responses to the study's findings.

4. Explore why the effects of COVID-19 on different types of cancer varied in this study.

6. PLOS authors have the option to publish the peer review history of their article (what does this mean?). If published, this will include your full peer review and any attached files.

Reviewer #1: No

Reviewer #2: No

---

## [Author Response · Author response to Decision Letter 0]

10 Dec 2023

PONE-D-23-21127

Impact of Coronavirus Disease 2019 on Cancer Care: How the Pandemic has Changed Cancer Utilization and Expenditures

Thank you for the opportunity to revise our manuscript. In this response letter below, two reviewers’ comments are italicized, and our point-by-point response to each comment is written in regular font. In our revised manuscript, we used red-colored texts to indicate where changes have been made. 

Reviewers' comments:

Reviewer #1: This study assessed the impact of COVID-19 on the diagnosis and management of patients with cancer. The authors found that the overall use of medical services by patients with cancer decreased in 2020 compared with that in the pre-COVID-19 pandemic period. This topic is interesting, and the manuscript is well-written. I just have several minor suggestions.

1. Add more detail data in the result part of abstract section and also result section.

Response: Thank you for your kind consideration about the description of the results. We have complemented the details in our Abstract which now reads: “Results: The incidence of cancer diagnoses has seen a notable decline since the outbreak of the COVID-19 in 2020 as compared to 2019. Despite the impact of COVID-19, there hasn't been a distinct decline in outpatient utilization when compared to inpatient utilization. While medical expenses for both inpatient and outpatient visits have slightly increased, the number of patients treated for cancer has decreased significantly compared to the previous year. In June 2020, overall outpatient costs experienced the highest increase (21.1%), while individual costs showed the most significant decrease (-4.9%) in June 2020. Finally, the number of hospitalisations and outpatient visits increased slightly from June–July in 2020, reducing the difference between the actual and predicted values. The decrease in the number of inpatient hospitalisations (-22~-6%) in 2020 was also high.”

In addition, we also updated the Results section which now reads: “Overall, the incidence of cancer-related insurance claims decreased during the COVID-19 pandemic. In particular, during the period of intensive social distancing in March and April of 2020 versus 2019, the incidence of gastric and colorectal cancers declined sharply, which seems to be related to the decreased cancer screening rate. In 2020, compared to the same month in 2019, the incidence of cancer decreased overall, excluding lip, oral, and pharyngeal cancer (1.9%), pancreatic cancer (0.3%), and testicular cancer (2.9%). Particularly, there was a significant decrease in thyroid cancer (-11.9%), stomach cancer (-11.1%), and brain and central nervous system cancers (-10.9%) (Figure 1). Regarding the changes in medical use related to hospitalisation, the number of hospitalisations decreased during the COVID-19 pandemic. In particular, the number of hospitalisations has continued to decline since the high-intensity social distancing policy was implemented (March–May) and during the second wave of the pandemic that occurred in a metropolitan area (August and September). In June 2020, outpatient visits (14.6%) showed a notable increase compared to the previous year, indicating a higher growth rate than in other months. Owing to the influence of COVID-19, the implementation of high-intensity social distancing seemed to have affected the number of outpatient visits, which decreased from April to May (ranging from -13.4% to -12.6%), and from August to October (ranging from -14.0% to -11.9%) (Figure 2). 

The relative percent changes in billing costs related to medical use in 2020 compared with that in 2019 are presented in Figure 3. After the initial outbreak of COVID-19 in 2020, the number of hospital admissions among cancer patients decreased overall compared with that in 2019 (ranging from -15.5% to -2.7%), although the total hospitalisation costs increased slightly. During the first, second, and third waves of the pandemic, the total costs from admissions decreased slightly, although the individual costs from admissions increased compared with that in 2019. Since the COVID-19 outbreak in 2020, the total costs related to outpatient visits increased compared with that in 2019. Individual costs from outpatient visits also increased overall compared with those in 2019, and the total outpatient costs increased the most in June (21.1%) when the number of confirmed COVID-19 cases stabilised; in contrast, individual costs from hospitalized patients decreased the most in June (-4.9%).

 After the first wave of the pandemic occurred in Daegu and Gyeongbuk from February–March 2020 in Korea, the proportion of patients hospitalised within 30 days of a cancer diagnosis showed the largest decrease (Figure 4). Breast (-3.9%), lung (-4.4%), and pancreatic (-4.6%) cancer patients showed the lowest decrease in the number of hospitalizations within 30 days compared to patients with other types of cancer. The reduction in the number of hospitalizations within 30 days for colorectal (-9.0%), stomach (-11.0%), and thyroid cancer (-11.7%) was particularly pronounced in July and August, which coincided with the period of the lowest social distancing measures due to the COVID-19 pandemic, compared to the same months in the previous year. Since then, the number of patients hospitalised within 30 days of a cancer diagnosis decreased in Seoul and Gyeonggi Province more than any other area owing to the second wave of the pandemic, which was centred in the Seoul metropolitan area, from July–August. Since November 2020, the number of patients with confirmed COVID-19 has increased to an average of 1,000 per day, and the number of inpatients nationwide has declined.

 The number of medical claims for hospitalisations of cancer-related diseases showed an increasing trend from 2014–2019, before the COVID-19 outbreak of in Korea. However, the number of medical claims in 2020 decreased by 8.7% to approximately 1.42 million (Supplementary Table S1). The number of admissions and outpatient visits gradually increased from 2014–2019; however, the number of admissions and outpatient has decreased since February 2020 (Figure 5). The actual number of treatments compared with the number of treatments predicted using the seasonal ARIMA model and, in particular, the decrease in the number of inpatient hospitalisations (-22~-6%) compared with the number of outpatient visits (-7%~-5%) was higher. Owing to the COVID-19 pandemic, the actual number of treatments decreased significantly compared with the predicted number of treatments, although there was no significant difference in medical costs. Both the number of admissions and outpatient visits showed the largest difference between the actual and predicted number of cases during the first wave of the pandemic (February–March). The number of hospitalisations and outpatient visits increased slightly from June–July, reducing the difference between the actual and predicted values.”

2. Please update the epidemiology of COVID-19 in the introduction section.

Response: Thanks for your comment. We have updated the epidemiology of COVID-19 in the first paragraph of Introduction section which now reads: “As of November 30, 2023, a total of 34,571,873 cumulative confirmed cases and 35,934 deaths have been reported in South Korea. Globally, it was reported to World Health Organization (WHO) that there are 772,052,752 confirmed cases of COVID-19 with 6,985,278 deaths [2].”

3. Please define how to calculate the cost and medical resource utilization of patients with cancer.

Response: Thank you. We have complemented the descriptions of calculation of the cost and medical resource utilization of cancer patients to the first paragraph under header "Outcome measures” of method section, which now reads: “This time-series analysis study identified whether changes in cancer care costs and utilization were affected by the COVID-19 pandemic. The NHIS claim database contains data on healthcare resource utilization, including disease diagnosis, medical treatment procedures, costs, and medication usage [19]. In our study, we calculated the resource utilization and costs, such as those related to administration, procedures, medication, injections, and other aspects of care, for the cancer population. We classified these into the categories of hospitalization and outpatient visit. Only the coverage by the National Health Insurance and co-payment with patients were considered. Specially for inpatient, the cost of admission was also included.”

4. May add statistical analysis to clarify the association between the trend and COVID-19.

Response: Thank you for your insightful comment. We acknowledge the importance of incorporating additional statistical analyses to elucidate the association between the observed trend and COVID-19. However, the delayed cancer registration announcements and persistent challenges in accessing NHIS data make it difficult to promptly present new statistical analyses. To assess the impact of COVID-19 in 2021, we anticipate that NHIS data will become available only after 2025. Given the urgency of our research topic, we utilized data from the National Health Insurance, covering the entire Korean population. We plan to address this as a limitation in our Discussion section, which currently reads: “In addition, presenting prompt statistical analyses to clarify the association between the observed trend and COVID-19 is challenging. Due to the urgency of our research, we relied on National Health Insurance data, encompassing the entire Korean population. However, persistent challenges in accessing NHIS data expected only after 2025, coupled with difficulties arising from delayed cancer registration announcements, make it difficult to conduct additional statistical analyses.”

5. Did this database include cancer stage? I wonder whether the effect of COVID-19 differ according to the cancer stage.

Response: Thank you for raising this question. It would indeed be significant to observe the impact of COVID-19 varying across different cancer stages. However, it's important to note that NHIS data lacks information on cancer stage. Consequently, analyzing the data stratified by cancer stage poses a challenge. We intend to address this limitation in our Discussion section, which currently reads: “Third, the NHIS data lacks information on cancer stage, investigating the impact of COVID-19 across different cancer stages poses a challenge.”

6. Please add the discussion about why the effect varied according to different cancer type.

Response: We appreciate your insightful feedback on discussion about why the effect varied according to different cancer type. We complemented the effects of COVID-19 on different types of cancer in our end of the 3rd, 5th and 6th paragraph of Discussion which now reads: “However, patients with pancreatic cancer had the lowest decrease in outpatient visits and hospitalizations compared to patients with other types of cancer. The study findings are justifiable, given the imperative nature of pancreatic cancer, which represents a major urgency in oncology and necessitates a substantial allocation of medical resources [37].”, “Indeed, especially during the period of high-intensity social distancing, there was a significant decrease in the incidence of stomach, colorectal, liver cancer, and breast cancer. Considering that stomach, liver, colorectal, breast, cervical, and lung cancers, can be screened through the Korean National Cancer Screening Program [40], it can be attributed largely to the significant decrease in cancer screening rates during the COVID-19 pandemic [5, 41].” and “In addition, outpatient medical expenses for pancreatic, lung, breast, liver, and other types of cancer increased compared to the same period the previous year. While in terms of inpatient costs, during the period of intensified social distancing measures, there was a significant decrease in inpatient costs for patients with most cancers, excluding pancreatic cancer. Furthermore, the government should consider how it may struggle to recover from the COVID-19 pandemic and establish permanent health policies for patients with cancer considering their cancer type and their vulnerability.”

Reviewer #2: This study investigated the impact of the COVID-19 pandemic on medical resource utilization and costs among cancer patients in South Korea. Using claims data, it observed a significant reduction in cancer diagnoses in 2020 compared to 2019, particularly in inpatient visits. While medical expenses remained stable, there was a noticeable decline in the number of cancer patients undergoing treatment. The study highlights the necessity for long-term health policies to address the pandemic's impact on cancer care and emphasizes the importance of future recovery efforts within the South Korean healthcare system for cancer patients.

Response: We appreciate your insightful feedback on our study examining the impact of the COVID-19 pandemic on medical resource utilization and costs among cancer patients in South Korea. We are grateful for the acknowledgment of the observed reduction in cancer diagnoses, especially with inpatient visits, in 2020 compared to 2019. The stability of medical expenses coupled with a noticeable decline in the number of cancer patients undergoing treatment is a crucial finding that we aim to emphasize. We will ensure that these important points are appropriately emphasized in the final version of our manuscript. We sincerely appreciate your valuable insights, which will undoubtedly contribute to the overall quality and relevance of our study.

Minor Suggestions:

1. Consider adding the cancer stage.

Response: Thank you for raising this question. It would indeed be significant to observe the impact of COVID-19 varying across different cancer stages. However, it's important to note that NHIS data lacks information on cancer stage. Consequently, analyzing the data stratified by cancer stage poses a challenge. We intend to address this limitation in our Discussion section, which currently reads: “Third, the NHIS data lacks information on cancer stage, investigating the impact of COVID-19 across different cancer stages poses a challenge.”

2. Include information about the outcomes of these cancer patients.

Response: Thank you. We have complemented the descriptions of calculation of the cost and medical resource utilization of cancer patients to the first paragraph under header "Outcome Measures” of method section, which now reads: “This time-series analysis study identified whether changes in cancer care costs and utilization were affected by the COVID-19 pandemic. The NHIS claim database contains data on healthcare resource utilization, including disease diagnosis, medical treatment procedures, costs, and medication usage [19]. In our study, we calculated the resource utilization and costs, such as those related to administration, procedures, medication, injections, and other aspects of care, for the cancer population. We classified these into the categories of hospitalization and outpatient visit. Only the coverage by the National Health Insurance and co-payment with patients were considered. Specially for inpatient, the cost of admission was also included.”

3. Discuss potential government responses to the study's findings.

Response: Thank you for this meaningful suggestion. We fully agree with the reviewer's recognition of the study's implications for long-term health policies. The observed trends underscore the urgency of addressing the pandemic's impact on cancer care and emphasize the need for strategic recovery efforts within the South Korean healthcare system, particularly to the challenges faced by cancer patients. Therefore, the Korean government is expected to discuss the final results of the termination of COVID-19, and it anticipates making statements on the outcomes regarding both COVID-19 and cancer after the official announcement of statistics on cancer registrations for the years 2022-2023, which is scheduled for the latter half of next year. Currently, South Korea is implementing ‘essential healthcare policies’ related to COVID-19 due to a shortage of specialized personnel. The primary goal of this essential healthcare is to emphasize the accessibility of medical services in residential areas. This has been particularly crucial during periods when the epidemic spread, affecting healthcare utilization, especially in certain regions, as evidenced by our results. Considering the significant impact on reducing healthcare utilization during the period when the epidemic spread, especially in some regions, and the occurrence of regional disparities in the spread of COVID-19 through community transmission after the first confirmed case in South Korea, it is believed that this policy will be very helpful. Therefore, we discussed this point in our 7th paragraph of Discussion which now reads: “The observed trends of our study results underscore the urgency of addressing the pandemic's impact on cancer care and emphasize the need for strategic recovery efforts within the South Korean healthcare system, particularly to the challenges faced by cancer patients. Therefore, it is anticipated that the South Korean government will soon discuss the final outcomes of efforts to terminate COVID-19. The government plans to make statements on both COVID-19 and cancer after the official announcement of cancer registration statistics for the years 2022-2023, scheduled for the latter half of next year. Currently, South Korea is implementing 'essential healthcare' policies in response to a shortage of specialized personnel for COVID-19. The primary goal of 'essential healthcare' is to emphasize the accessibility of medical services in residential areas [49, 50]. This emphasis has been particularly crucial in regions where healthcare utilization was significantly impacted during the period of epidemic spread, especially evident in certain areas, consistent with our results. Taking into account the substantial decrease in healthcare utilization during the epidemic's spread, particularly in certain regions, and the emergence of regional disparities in COVID-19, it is expected that this policy will prove highly beneficial.”

4. Explore why the effects of COVID-19 on different types of cancer varied in this study.

Response: We appreciate your insightful feedback on discussion about why the effect varied according to different cancer type. We complemented the effects of COVID-19 on different types of cancer in our end of the 3rd, 5th and 6th paragraph of Discussion which now reads: “However, patients with pancreatic cancer had the lowest decrease in outpatient visits and hospitalizations compared to patients with other types of cancer. The study findings are justifiable, given the imperative nature of pancreatic cancer, which represents a major urgency in oncology and necessitates a substantial allocation of medical resources [37].”, “. Indeed, especially during the period of high-intensity social distancing, there was a significant decrease in the incidence of stomach, colorectal, liver cancer, and breast cancer. Considering that stomach, liver, colorectal, breast, cervical, and lung cancers, can be screened through the Korean National Cancer Screening Program [40], it can be attributed largely to the significant decrease in cancer screening rates during the COVID-19 pandemic [5, 41].”, and “In addition, outpatient medical expenses for pancreatic, lung, breast, liver, and other types of cancer increased compared to the same period the previous year. While in terms of inpatient costs, during the period of intensified social distancing measures, there was a significant decrease in inpatient costs for patients with most cancers, excluding pancreatic cancer. Furthermore, the government should consider how it may struggle to recover from the COVID-19 pandemic and establish permanent health policies for patients with cancer considering their cancer type and their vulnerability.”

---

## [Decision Letter · Decision Letter 1]

19 Dec 2023

Impact of Coronavirus Disease 2019 on Cancer Care: How the Pandemic has Changed Cancer Utilization and Expenditures

PONE-D-23-21127R1

Dear Dr. Kim,

We’re pleased to inform you that your manuscript has been judged scientifically suitable for publication and will be formally accepted for publication once it meets all outstanding technical requirements.

Kind regards,

Chong-Chi Chiu

Academic Editor

PLOS ONE

---

## [Editor Report · Acceptance letter]

29 Jan 2024

PONE-D-23-21127R1 

PLOS ONE

Dear Dr. Kim, 

I'm pleased to inform you that your manuscript has been deemed suitable for publication in PLOS ONE. Congratulations! Your manuscript is now being handed over to our production team.

Kind regards, 

on behalf of

Professor Chong-Chi Chiu 

Academic Editor

PLOS ONE